# An Alternative Treatment Effect Measure for Time-to-Event Oncology Randomized Trials

**DOI:** 10.3390/cancers17233750

**Published:** 2025-11-24

**Authors:** Alan D. Hutson, Han Yu

**Affiliations:** Roswell Park Comprehensive Cancer Center, Department of Biostatistics and Bioinformatics, Elm and Carlton Streets, Buffalo, NY 14623, USA

**Keywords:** survival analysis, martingale residuals, randomization test, proportional hazards, permutation test

## Abstract

This work introduces a new survival analysis measure called the univariate martingale residual (UMR) for comparing treatments in phase III oncology trials. Traditional methods like hazard ratios and log-rank tests often rely on assumptions such as proportional hazards and large sample sizes, which may not hold in complex trial settings. The UMR provides an unbiased summary of treatment effects by quantifying the average difference between observed and expected events. It also enables exact statistical testing without relying on model-based assumptions. Overall, the UMR offers a more robust and interpretable alternative for evaluating treatment efficacy in challenging survival analysis scenarios.

## 1. Introduction

In phase III oncology trials, time-to-event endpoints are essential for evaluating treatment efficacy. The most common endpoints include Overall Survival (OS), Progression-Free Survival (PFS), and Event-Free Survival (EFS). OS, defined as the time from randomization to death from any cause, is considered the gold standard due to its objectivity, although it typically requires long follow-up and can be influenced by subsequent therapies. PFS measures the time to disease progression or death and is widely used, especially in metastatic settings, though it depends on consistent imaging schedules and standardized progression criteria. EFS, which captures the time to predefined events such as progression, recurrence, or death, is frequently used in early-stage cancers or adjuvant settings, such as in leukemia or lymphoma.

Additional commonly used endpoints include Disease-Free Survival (DFS), which focuses on recurrence or death after curative treatment; Time to Progression (TTP), which measures progression while excluding non-cancer-related deaths; Time to Treatment Failure (TTF), which captures treatment discontinuation due to progression, toxicity, or other reasons; and Duration of Response (DoR), which quantifies the duration of tumor response in patients who initially respond to treatment.

Endpoint selection depends on cancer type, disease stage, trial objectives, and regulatory requirements. OS and PFS are most commonly prioritized due to their clinical relevance and regulatory acceptance, while EFS and DFS are prevalent in early-stage or adjuvant trials. TTP, TTF, and DoR typically serve as secondary endpoints.

In this note, we focus on straightforward two-group comparisons in randomized phase III trials. Without loss of generality to more complex designs, we introduce a new endpoint termed the univariate martingale residual for comparing survival curves that facilitates both nonparametric estimation of treatment effects per arm and exact Type I error control within the framework of a randomization test. The methods discussed are also applicable to designs that incorporate stratification or to earlier-phase trials.

For time-to-event endpoints in two-group comparisons, the primary statistical measure of treatment effect is the hazard ratio. This is typically estimated using the Cox proportional hazards model and quantifies the relative risk of an event between treatment arms under the proportional hazards assumption. These estimates are usually accompanied by 95% confidence intervals to reflect estimation precision. The log-rank test is commonly used to compare survival curves and assess the statistical significance of differences in event times over the follow-up period. In the two-group setting, the log-rank test and the score test from the Cox model yield equivalent *p*-values under the null hypothesis of equal hazards [1].

Median survival times are derived from Kaplan–Meier (KM) estimates and indicate the time by which 50% of participants have experienced the event. These are also reported with 95% confidence intervals. Event-free probabilities at fixed time points, also derived from KM curves, provide clinically interpretable measures of the proportion of patients who remain event-free at specified intervals.

The Cox model and the log-rank test for comparing treatment arms both rely on the proportional hazards assumption. While this assumption does not affect Type I error control in randomized trials under the null hypothesis, provided that asymptotic approximations are accurate, it can lead to substantial power loss under alternatives. In particular, when hazard rates over time between the two groups are not proportional, the hazard ratio may provide a biased and potentially misleading summary of treatment effect at any specific time point. Although these methods are powerful when the assumptions hold, they depend heavily on large-sample approximations.

Several studies have reported inflated Type I error rates associated with the Cox model in small-sample settings [2,3,4]. As Shao et al. [4] observe, the Cox model often performs poorly when the number of events is small or when there is heavy censoring in one or both treatment groups. In response to these limitations, Shao et al. proposed an exact test based on the Cox model. However, this test still assumes proportional hazards to achieve optimal power and to support unbiased estimation of the hazard ratio.

Alternative point estimates, such as median survival time, may be undefined in the presence of censoring. Even when defined, they may fail to capture important differences in survival if the curves overlap at the median despite diverging elsewhere. Similarly, survival probabilities at fixed time points may miss broader differences between groups if the curves cross at those specific times.

In order to get a sense of what is being used in phase III oncology trials we examined a series of 30 phase III oncology trials published sequentially in the *New England Journal of Medicine* from 2023–2025 [5,6,7,8,9,10,11,12,13,14,15,16,17,18,19,20,21,22,23,24,25,26,27,28,29,30,31,32,33,34]. Across the 30 phase III clinical trials, survival rates at specific time points were reported as follows: no trials (0%) reported 6-month survival rates, 2 trials (6.6%) reported 1-year rates for progression-free survival (PFS) or treatment failure-free survival, 1 trial (3.3%) reported 18-month event-free survival (EFS), and 7 trials (23.3%) reported 2-year rates for PFS, EFS, disease-free survival (DFS), or overall survival (OS), making it the most common time point. Additionally, 2 trials (6.6%) reported 3-year rates for PFS, OS, or invasive disease-free survival (IDFS), 1 trial (3.3%) reported 5-year DFS, 1 trial (3.3%) reported 8-year OS, and 1 trial (3.3%) reported 10-year OS and local recurrence rates. All survival probabilities and median survival times (e.g., median PFS, OS, EFS, or DFS) were calculated using the Kaplan–Meier method, with hazard ratios (HRs) estimated by stratified Cox proportional-hazards models in all 30 trials (100%) to compare treatment groups, often stratified by factors like mutation status or disease stage. The log-rank test was used in 29 trials (96.7%) to compare survival distributions, often stratified, with two trials focusing on noninferiority analyses. The 2-year rates were most frequently reported due to their relevance in assessing treatment efficacy in advanced or metastatic cancers, while longer-term rates (5–10 years) were less common, reflecting shorter median follow-up periods (7.8–62 months) in most trials.

In this note, we introduce and evaluate a new metric, termed the **univariate martingale residual** (UMR) [35], considering both its use as a test statistic and as a summary measure of treatment effect within each treatment arm. In essence, the martingale residual quantifies the difference between the observed and expected number of events. Subject-level UMRs can be grouped by treatment arm in a randomized study under the null hypothesis of no treatment effect, and comparisons may be made by examining the average UMRs across arms. Further details are provided in the next section.

In this note, we outline the development and application of the UMR. Section 2 introduces its formal definition. Section 3 illustrates the randomization test based on UMRs in a two-group setting. In Section 4, we present simulated examples and a corresponding power study. Section 5 provides three real-world applications, followed by concluding remarks in Section 6.

## 2. Definition of Univariate Martingale Residuals

We briefly review a recently proposed randomization test framework for right-censored outcomes, originally developed by Hutson and Yu [35], which integrates a martingale-residual-based approach into the classical randomization test setting. This method allows for exact Type I error control and is especially advantageous when standard assumptions such as proportional hazards are violated or when the sample size is small.

In the simple parallel-group design, the randomization test proceeds by considering the outcome variable of interest, in this case, the univariate martingale residual (UMR) defined below, under the strict additivity framework of Kempthorne [36]. Suppose N=n+m subjects are randomized to treatments *A* and *B*, respectively, with all Nn assignments equally likely. Let $t_{1},t_{2},\ldots ,t_{N}$ represent independent and identically distributed failure times, and let $c_{1},c_{2},\ldots ,c_{N}$ denote the corresponding non-informative right censoring times, independently distributed and indexed by i=1,2,…,N, pooled across treatments *A* and *B*. Due to right censoring, we only observe g≤N of the *t*’s, where $y_{i}=\min\left(t_{i},c_{i}\right)$ and $\delta_{i}=I_{\left(t_{i}>c_{i}\right)}$. The UMR for subject *i*, i=1,2,…,N, is defined as:(1)$$\begin{array}{c} r_{i}=\delta_{i}-\hat{H}\left(y_{i}\right), \end{array}$$
where $\delta_{i}$ is the event indicator and $\hat{H}\left(y_{i}\right)=-\log\left(\hat{S}\left(y_{i}\right)\right)$ is the estimated cumulative hazard at time $t_{i}$, obtained from the product-limit estimator of the survival function. This estimator, derived from discretized hazard functions, follows the classic form:(2)$$\hat{S}\left(y\right)=\prod_{y_{\left(j\right)}\leq y}\left(1-{\hat{h}}_{j}\right),$$
with ${\hat{h}}_{j}=d_{j}/r_{j}$, where $d_{j}$ and $r_{j}$ are the number of events and number at risk at time $y_{\left(j\right)}$, respectively [37].

Martingale residuals, originally discussed in the context of model diagnostics for the Cox regression model [38], have the appealing property that $\sum_{i=1}^{N}r_{i}=0$. In the randomization test framework, these residuals can be treated as fixed outcomes [39], with randomness only in the treatment assignment. In the univariate setting the UMR’s also exhibit the property $\sum_{i=1}^{N}r_{i}=0$ [38].

The UMR, as defined at (Equation 1), can be interpreted at each observed $y_{i}$ as the expected number of excess deaths over [0,t]. When the UMRs are delineated by treatment arm we would expect a poorly performing treatment to have a large proportion of observed $r_{i}$’s greater than zero and a efficacious treatment to have a large proportion of observed $r_{i}$’s less than zero given the pooled sample. The observed UMR values form the basis for constructing an exact test comparing treatment arms An advantage of using UMRs, as noted in Hutson and Yu [35], is that they remain estimable even when median survival times are undefined due to heavy censoring. Their mean and standard deviation can be directly computed by treament arm, offering interpretable summary statistics that retain efficiency and robustness across a variety of censoring scenarios. UMRs can be easily output using standard statistical software pacakges for fitting a Cox regression model by simply using a dummy variable covariate set to one and outputting the martingale residuals in a standard fashion.

## 3. Randomization Testing with UMRs

The null hypothesis for comparing treatment arms *A* and *B* using UMRs can be written as$$H_{0}:F_{A}\left(s\right)=F_{B}\left(s\right),\;\forall s,$$
where $F_{A}\left(s\right)$ and $F_{B}\left(s\right)$ are the empirical cumulative distribution functions (c.d.f.s) of the UMRs given fixed observations for treatments *A* and *B*, respectively. Standard test statistics such as the Wilcoxon rank-sum or a randomization *t*-statistic may be used. In this work, we employ the difference in average Martingale residuals between groups *A* and *B*.

In the parallel-group setting, the randomization test proceeds as follows. For any outcome variable, discrete, ordinal, or continuous, an exact randomization test is applicable. Here, the outcome variable is the UMR. Let N=n+m, where *n* and *m* denote the numbers assigned to treatments *A* and *B*, respectively. There are Nn equally likely assignments, each with probability 1/Nn.

Let $I_{i}\in \left\{0,1\right\}$ denote the treatment indicator for subject *i*(i=1,…,N), with $P\left(I_{i}=1\right)=\pi \in \left(0,1\right)$. Following Rosenbaum [39], the observed UMR is$$R_{i}=I_{i}r_{A_{i}}+\left(1-I_{i}\right)r_{B_{i}},$$
where $I_{i}=1$ indicates treatment *A* and $I_{i}=0$ indicates treatment *B*. Under strict additivity [36], the $\left\{r_{i}\right\}$ are fixed, with randomness arising solely from the $\left\{I_{i}\right\}$.

We define $r_{A_{i}}=s_{i}+a_{i}$ and $r_{B_{i}}=s_{i}+b_{i}$, where $s_{i}$ is the subject effect and $a_{i},b_{i}$ the treatment effects. Thus, if both treatments were hypothetically applied to subject *i*, the difference $r_{A_{i}}-r_{B_{i}}=a_{i}-b_{i}$ would be constant across subjects. Inference is therefore driven by randomization.

For one-sided testing,$$H_{0}:F_{A}\left(s\right)=F_{B}\left(s\right),\;\forall s\;\mathrm{vs}.\;H_{1}:F_{A}\left(s\right)>F_{B}\left(s\right),\;\forall s,$$ (or analogously $H_{1}:F_{A}\left(s\right)<F_{B}\left(s\right)$). Under a shift alternative, where both c.d.f.s belong to the same family, this is equivalent to testing about the expectations and expressed as$$H_{0}:E\left[R|I=1\right]=E\left[R|I=0\right]\;\mathrm{vs}.\;H_{1}:E\left[R|I=1\right]>E\left[R|I=0\right].$$

Define the observed test statistic$$S={\bar{r}}_{A}-{\bar{r}}_{B},\;{\bar{r}}_{A}=\frac{1}{n}\sum_{j=1}^{N}I_{j}R_{j},\;{\bar{r}}_{B}=\frac{1}{m}\sum_{j=1}^{N}\left(1-I_{j}\right)R_{j}.$$

For each of the l=Nn permutations, compute$$S_{k}={\bar{r}}_{A,k}-{\bar{r}}_{B,k},\;k=1,\ldots ,l.$$

The exact *p*-values are$$p=\frac{1}{l}\sum_{k=1}^{l}1\left\{S_{k}\geq S\right\},\;\mathrm{for}\;H_{1}:E\left[R|I=1\right]>E\left[R|I=0\right],$$$$p=\frac{1}{l}\sum_{k=1}^{l}1\left\{S_{k}\leq S\right\},\;\mathrm{for}\;H_{1}:E\left[R|I=1\right]<E\left[R|I=0\right],$$
where 1{·} is the indicator function.

For a two-sided alternative,$$H_{1}:E\left[R|I=1\right]\neq E\left[R|I=0\right],$$
a natural test statistic is$$S=|{\bar{r}}_{A}-{\bar{r}}_{B}|,$$
with *p*-value$$p=\frac{1}{l}\sum_{k=1}^{l}1\left\{S_{k}^{2}\geq S^{2}\right\}.$$

In practice, *p*-values are often estimated via *B* Monte Carlo permutations, replacing *l* with *B*. Typically B=10,000 yields stable estimates.

**Type I Error Control.** Each configuration of the UMR vector r has probability 1/Nn. Let $S^{*}$ denote the test statistic under a re-randomization. We test$$H_{0}:F_{A}\left(s\right)=F_{B}\left(s\right),\;\forall s$$
against either$$H_{1}:F_{A}\left(s\right)>F_{B}\left(s\right),\;\forall s,\;\mathrm{or}\;\mathrm{in}\;\mathrm{the}\;\mathrm{location}\text{-}\mathrm{shift}\;\mathrm{model}\;H_{1}:E\left[R|I=1\right]>E\left[R|I=0\right].$$

The critical value $t_{\alpha }$ is chosen such that$$\underset{H_{0}}{\mathrm{Pr}}\left(S^{*}\geq t_{\alpha }\right)\leq \alpha .$$

Let $s_{\alpha }$ be the threshold yielding a significance level as close to α as possible. Without ties, the exact level is⌊αNn⌋Nn,
where ⌊·⌋ is the integer part. With ties near $t_{\alpha }$, the level may be slightly lower.

**Comment on Blocking and Stratification.** If blocked randomization or stratification is used, the test is modified by permuting treatment assignments within blocks and/or strata.

## 4. Simulation Study

### 4.1. Examples

To illustrate the proposed methods by mimicking realistic settings, we consider several simulation scenarios based on Weibull-distributed random variables. Let $Y_{B}$ and $Y_{A}$ denote random variables with quantile functions$$Q_{B}\left(u\right)={\left[-\sigma_{B}\log\left(1-u\right)\right]}^{\theta_{B}},\;Q_{A}\left(u\right)={\left[-\sigma_{A}\log\left(1-u\right)\right]}^{\theta_{A}},$$
where $\sigma_{B},\sigma_{A}>0$ and $\theta_{B},\theta_{A}>0$. All tests used in the examples were two-sided.


**Hazard Functions**


For a Weibull quantile function $Q\left(u\right)={\left[-\sigma \log\left(1-u\right)\right]}^{\theta }$, the corresponding hazard function is$$h\left(y\right)=\frac{1}{\theta \sigma }\;y^{\frac{1-\theta }{\theta }},\;y>0.$$

Thus:For $Y_{B}$: $h_{B}\left(y\right)=\frac{1}{\theta_{B}\sigma_{B}}\;y^{\frac{1-\theta_{B}}{\theta_{B}}}$,For $Y_{A}$: $h_{A}\left(y\right)=\frac{1}{\theta_{A}\sigma_{A}}\;y^{\frac{1-\theta_{A}}{\theta_{A}}}$.


**Hazard Ratio**


The hazard ratio of $Y_{B}$ relative to $Y_{A}$ is$$\frac{h_{B}\left(y\right)}{h_{A}\left(y\right)}=\frac{\theta_{A}\sigma_{A}}{\theta_{B}\sigma_{B}}\;y^{\frac{1-\theta_{B}}{\theta_{B}}-\frac{1-\theta_{A}}{\theta_{A}}}.$$

This ratio is constant only when $\theta_{B}=\theta_{A}$, in which case it simplifies to $\frac{\sigma_{A}}{\sigma_{B}}$.

Censoring is introduced via an exponential random variable *C* with quantile function$$Q_{C}\left(u\right)=-\sigma_{C}\log\left(1-u\right).$$


**Scenario 1**


We consider n=m=75 with parameters:$$\sigma_{0}=2,\;\theta_{0}=1,\;\sigma_{1}=1,\;\theta_{1}=1,\;\sigma_{C}=1.$$

Results.

**Group means of martingale residuals: **A=−0.142, B=0.142.**Randomization test (UMR difference): **p=0.0104.**Median survival: **1.096 for A, 0.615 for B.**Log-rank test: **$\chi^{2}=6.389$, p=0.0115.**Cox PH model: **$\hat{HR}=1.809$ with 95% CI (1.135,2.882), p=0.0127.

The Kaplan–Meier curves are shown in Figure 1. Under proportional hazards, the three tests yield consistent inference. The UMRs suggest that Arm A experienced an average of −0.142 excess deaths over [0,t] while Arm B experienced an average of 0.142, providing a clear interpretation of treatment differences. Inference across the log-rank, Cox Model and randomization test provide similar results.


**Scenario 2**


We consider n=m=75 with parameters:$$\sigma_{0}=2,\;\theta_{0}=\frac{1}{2},\;\sigma_{1}=1,\;\theta_{1}=1,\;\sigma_{C}=1.$$

Results.

**Group means of martingale residuals: **A=−0.117, B=0.117.**Randomization test (UMR difference): **p=0.0285.**Median survival: **1.121 for A, 0.977 for B.**Log-rank test: **$\chi^{2}=4.780$, p=0.0288.**Cox PH model: **$\hat{HR}=1.725$ with 95% CI (1.052,2.829), p=0.0307.

One-year survival probabilities are estimated as ${\hat{S}}_{A}\left(1\right)=0.54$ and ${\hat{S}}_{B}\left(1\right)=0.48$. The Kaplan–Meier curves (Figure 2) reveal clear non-proportional hazards as expected based on the simulation parameters. Median survival and fixed-time probabilities provide limited insight, while the UMRs again yield a simple and interpretable summary.


**Scenario 3**


We consider n=m=75 with parameters:$$\sigma_{0}=3,\;\theta_{0}=\frac{1}{2},\;\sigma_{1}=1,\;\theta_{1}=1,\;\sigma_{C}=\frac{1}{4}.$$

Results.

**Group means of martingale residuals: **A=−0.089, B=0.089.**Randomization test (UMR difference): **p=0.0007.**Median survival: ** not estimable for either group due to heavy censoring.**Log-rank test: **$\chi^{2}=10.783$, p=0.0010.**Cox PH model: **$\hat{HR}=8.191$ with 95% CI (1.853,36.208), p=0.0055.

The Kaplan–Meier curves (Figure 3) confirm that median survival and fixed-time survival probabilities are uninformative. Non-proportional hazards lead to a biased HR estimate, while UMRs provide an interpretable and stable summary: −0.089 excess deaths for A versus 0.089 for B.


**Scenario 4**


We consider n=m=75 with parameters:$$\sigma_{0}=3000,\;\theta_{0}=1,\;\sigma_{1}=1,\;\theta_{1}=1,\;\sigma_{C}=1.$$

Results.

**Group means of martingale residuals: **A=−0.298, B=0.298.**Randomization test (UMR difference): **p<0.0001.**Median survival: ** not estimable for A, 0.977 for B.**Log-rank test: **$\chi^{2}=53.562$, p<0.0001.**Cox PH model: **$\hat{HR}\approx 9.08\times 10^{8}$ with 95% CI (0,∞), p=0.996.

Here, no events occur in Arm A, a situation often encountered in practice. The Kaplan–Meier curves (Figure 4) show that the HR is not estimable, and the Cox model *p*-value is unreliable due to quasi-separation. In contrast, UMRs again yield a stable and interpretable measure: −0.298 excess deaths for A versus 0.298 for B, with inference consistent with the log-rank test.

Across all scenarios, UMRs provide a consistent summary measure and enable exact randomization inference, even when proportional hazards are violated or Cox models suffer from numerical instability. The next section evaluates type I error control and power.

### 4.2. Power Comparison

To assess the operating characteristics of the proposed randomization test, we conducted a simulation study under a variety of parameter settings. In each replicate, survival times were generated from Weibull-type models with varying scale parameters $\left(\sigma_{0},\sigma_{c}\right)$ and shape parameter $\theta_{0}$, while the comparison group was fixed at $\sigma_{1}=1$ and $\theta_{1}=1$. Each simulated dataset consisted of n=m=100 subjects per group, and censoring times were drawn to reflect different censoring distributions. For each scenario, N=1000 replicates were performed, and the randomization test was carried out with 1000 resamples. Both the log-rank test and the randomization test based on differences in average martingale residuals were evaluated at a significance level of α=0.05 (two-sided).

In general, the simulation results show that the log-rank test and the randomization test produce nearly identical rejection probabilities across all parameter combinations considered. This suggests that, in practice, the two methods perform equivalently, and thus an extensive simulation study is unnecessary beyond illustrative examples. As seen in Table 1 and Table 2, the estimated powers of the two procedures are consistently aligned, confirming that the permutation approach retains the operating characteristics of the traditional log-rank test while offering a useful alternative formulation. From a heuristic perspective, these findings further suggest that average UMRs may serve as an appropriate summary measure to pair with the log-rank test, or that the randomization test alone may be employed as a practical substitute.

## 5. Real World Examples

### 5.1. Example 1

Our first real-world example is based on a randomized trial comparing androgen-deprivation therapy (ADT) plus docetaxel versus ADT alone in patients with metastatic hormone-sensitive prostate cancer [40]. In this large study of 790 patients, the primary objective was to test the hypothesis that the median overall survival would be 33.3% longer among those receiving early docetaxel in addition to ADT compared with ADT alone. The sample sizes were n=397 for the ADT + docetaxel group (I=1) and m=393 for the ADT-alone group (I=0).

For illustration purposes, the original stratification factors were not used in our analysis. The results are summarized below, and the Kaplan–Meier survival curves for each treatment arm and combined are shown in Figure 5. The permutation UMR test was significant at α=0.05 (two-sided), indicating that patients receiving ADT alone experienced an average expected excess of 0.067 deaths over the study period, while those receiving ADT + docetaxel experienced an expected average decrease of −0.055 deaths. The inference from the log-rank test was consistent with the permutation UMR result. Visual inspection of the survival curves suggests non-proportional hazards, as the curves overlap during the early time period, indicating that the hazard ratio may not provide a fully adequate summary of treatment efficacy.

Results.

**Group means of martingale residuals: **I=0:0.067, I=1:−0.055.**Randomization test (UMR difference): ** two-sided p=0.0006.**Median survival: **44.0 for z=0, 57.6 for z=1.**Log-rank test: **$\chi^{2}=11.31$, p=0.00077.**Cox PH model: **$\hat{HR}=0.664$ with 95% CI (0.520,0.846), p=0.00095.

### 5.2. Example 2

For our second example, we utilized publicly available data from Project Data Sphere (https://data.projectdatasphere.org/) accessed on July 17, 2020. Specifically, we examined a randomized phase II clinical trial [41] that evaluated the efficacy and safety of LY2510924 (LY) in combination with first-line standard-of-care (SOC) chemotherapy for patients with extensive-disease small cell lung cancer (ED-SCLC). The primary efficacy endpoint of the original study was progression-free survival (PFS), with overall survival (OS) designated as a key secondary endpoint. For this illustration, we focused on OS.

The treatment groups were defined as follows: LY + SOC (Arm A; n=47, I=1) and SOC alone (Arm B; m=42, I=0). The results are summarized below, and the Kaplan–Meier survival curves for each treatment arm and the combined data are presented in Figure 6. The permutation UMR test was not significant at α=0.05 (two-sided). Descriptively, patients receiving LY + SOC experienced an average expected excess of 0.130 deaths over the study period, whereas those receiving SOC experienced an expected an average decrease of −0.146 deaths. The inference from the log-rank test was consistent with the permutation UMR test.

Visual inspection of the survival curves suggests clear evidence of non-proportional hazards, as the curves crisscross over time, indicating that the hazard ratio may not serve as an appropriate summary of treatment effect. Similarly, the estimated survival probabilities at six months were comparable between groups: ${\hat{S}}_{0}\left(6\right)=0.762$ and ${\hat{S}}_{1}\left(6\right)=0.801$.

Results.

**Group means of martingale residuals: **I=0:−0.146, I=1:0.130.**Randomization test (UMR difference): ** two-sided p=0.121.**Median survival: **11.1 for z=0, 9.7 for z=1.**Log-rank test: **$\chi^{2}=2.53$, p=0.112.**Cox PH model: **$\hat{HR}=1.52$ with 95% CI (0.905,2.540), p=0.114.

### 5.3. Example 3

Our third example is based on a randomized clinical trial involving post-surgical patients with Stage I–IIA cervical cancer who exhibited pathologically confirmed intermediate-risk factors, including combinations of capillary lymphatic space (CLS) involvement, stromal invasion, and tumor size [42]. Patients were randomly assigned in a 1:1 ratio to receive either adjuvant chemoradiotherapy (CRT; n=158) or radiotherapy (RT; m=158). All patients received conformal RT or intensity-modulated radiation therapy (IMRT). In the CRT arm, cisplatin was administered at a dose of 40 mg/m^2^ weekly for six cycles during RT. Recurrence-free survival (RFS) was the primary endpoint among randomized and eligible patients.

Because the raw data were not publicly available, we reconstructed a synthetic dataset with equivalent sample size and censoring distribution using the published Kaplan–Meier curves and the WebPlotDigitizer tool v5 (https://automeris.io/wpd/). These reconstructed data are used solely for illustrative purposes. The treatment groups were defined as CRT (I=0) and RT (I=1).

The results are summarized below, and the Kaplan–Meier survival curves for RFS by treatment arm and combined data are shown in Figure 7. As illustrated in Figure 7, the median RFS was not estimable for either treatment arm due to the degree of censoring. The permutation UMR test was not significant at α=0.05 (two-sided). Descriptively, patients receiving CRT experienced an average expected excess of 0.04 deaths over the study period, whereas those receiving RT experienced an average expected decrease of −0.04 deaths. The inference from the log-rank test was consistent with the permutation UMR result.

Results.

**Group means of martingale residuals: **I=0:0.040, I=1:−0.040.**Randomization test (mean difference): ** two-sided p=0.095.**Median survival: ** not estimable for either group.**Log-rank test: **$\chi^{2}=2.82$, p=0.093.**Cox PH model: **$\hat{HR}=0.64$ with 95% CI (0.378,1.082), p=0.096.

## 6. Conclusions

This paper introduces a direct and interpretable approach for quantifying treatment effects in survival analysis. The proposed measure, the *average univariate martingale residual* (UMR) per treatment arm, serves as a simple yet powerful summary statistic that represents the average number of excess deaths attributable to treatment differences. Unlike the conventional hazard ratio, which depends on the proportional hazards assumption, the UMR provides an assumption-free, unbiased estimate of treatment efficacy over the entire time course that can be interpreted on an absolute scale.

The method demonstrates robustness in complex and challenging trial settings. In particular, it yields stable and meaningful summaries under heavy censoring, where median survival cannot be reliably estimated, and in cases of quasi-complete separation, where the Cox model and associated hazard ratio estimates become unstable or undefined. These properties make the approach particularly valuable for modern oncology trials and other studies where such data complications are common.

In addition to introducing a new summary metric, we integrate the UMR within an exact randomization (permutation) testing framework to enable valid statistical inference. This combination allows the computation of exact *p*-values without reliance on large-sample approximations or model-based assumptions, in contrast to standard methods such as the log-rank test or Cox model. Collectively, this framework provides a rigorous and interpretable alternative for evaluating treatment effects in randomized clinical trials.

## Figures and Tables

**Figure 1 cancers-17-03750-f001:**
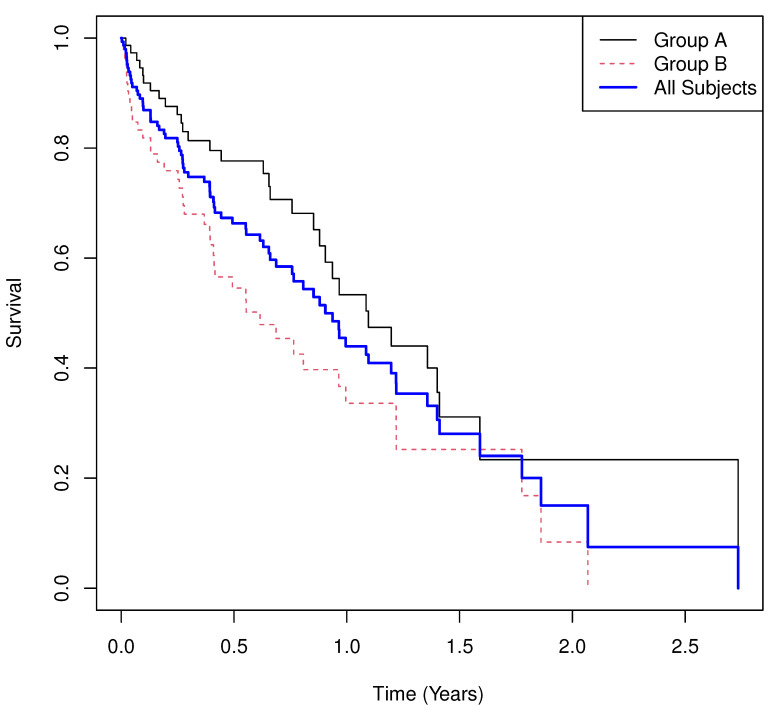
Simulated Example 1: n=m=75, $\sigma_{A}=2$, $\theta_{A}=1$, $\sigma_{B}=1$, $\theta_{B}=1$, and $\sigma_{C}=1$.

**Figure 2 cancers-17-03750-f002:**
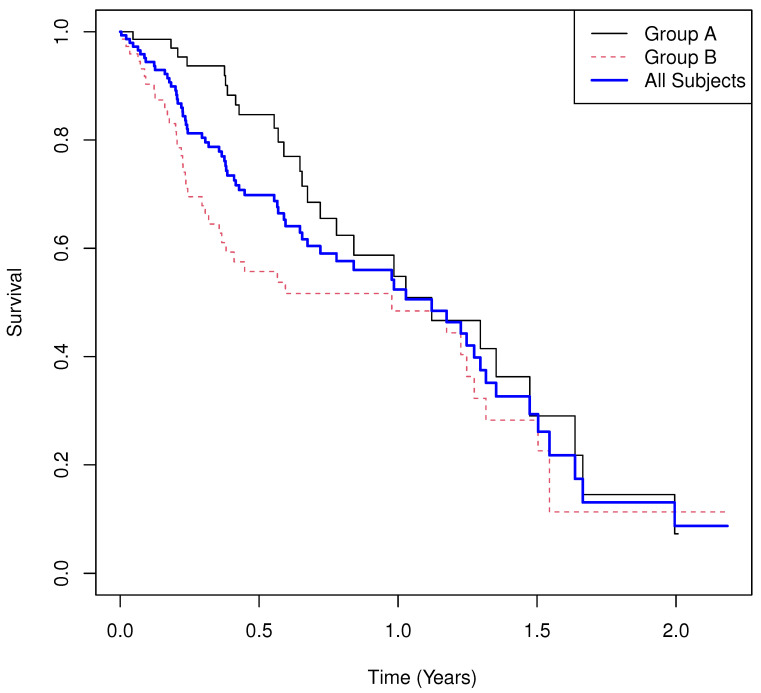
Simulated Example 2: n=m=75, $\sigma_{A}=2$, $\theta_{A}=1/2$, $\sigma_{B}=1$, $\theta_{B}=1$, and $\sigma_{C}=1$.

**Figure 3 cancers-17-03750-f003:**
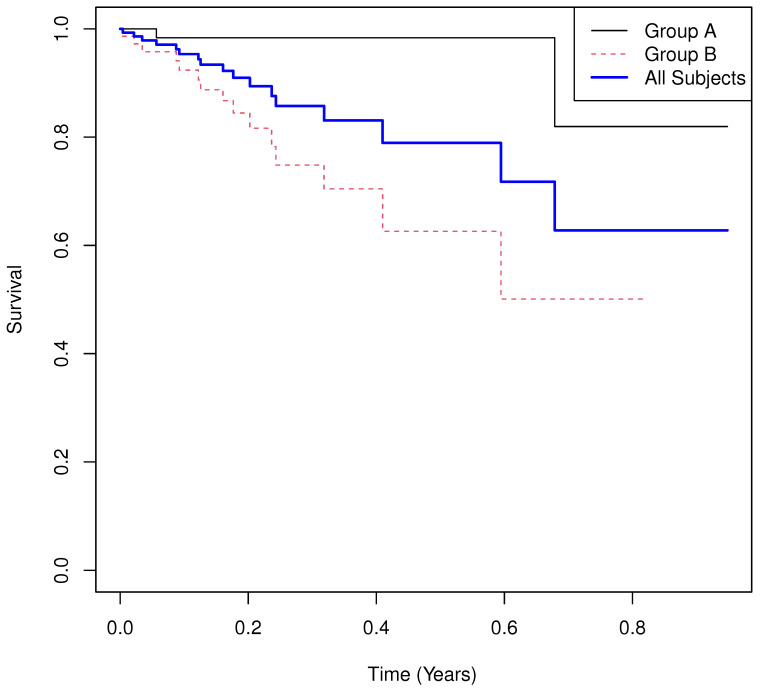
Simulated Example 3: n=m=75, $\sigma_{A}=3$, $\theta_{A}=1/2$, $\sigma_{B}=1$, $\theta_{B}=1$, and $\sigma_{C}=1/4$.

**Figure 4 cancers-17-03750-f004:**
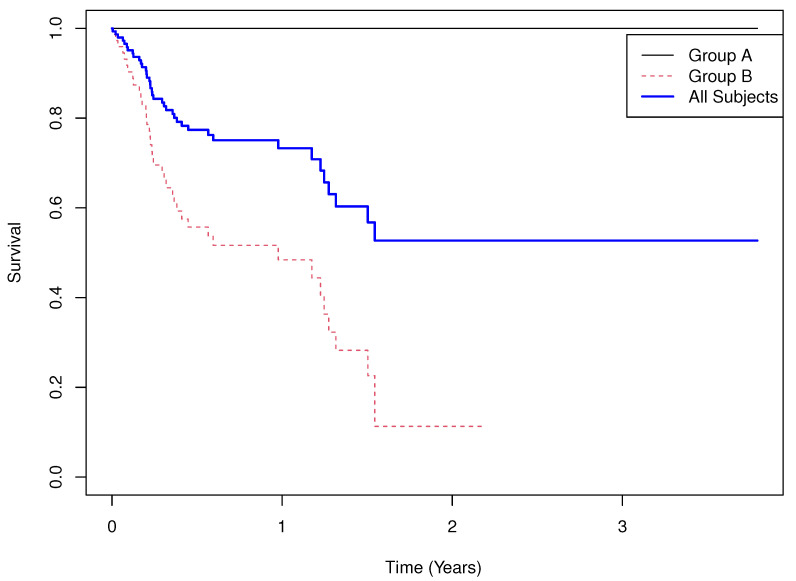
Simluated Example 4: n=m=75, $\sigma_{A}=3000$, $\theta_{A}=1$, $\sigma_{B}=1$, $\theta_{B}=1$, and $\sigma_{C}=1$.

**Figure 5 cancers-17-03750-f005:**
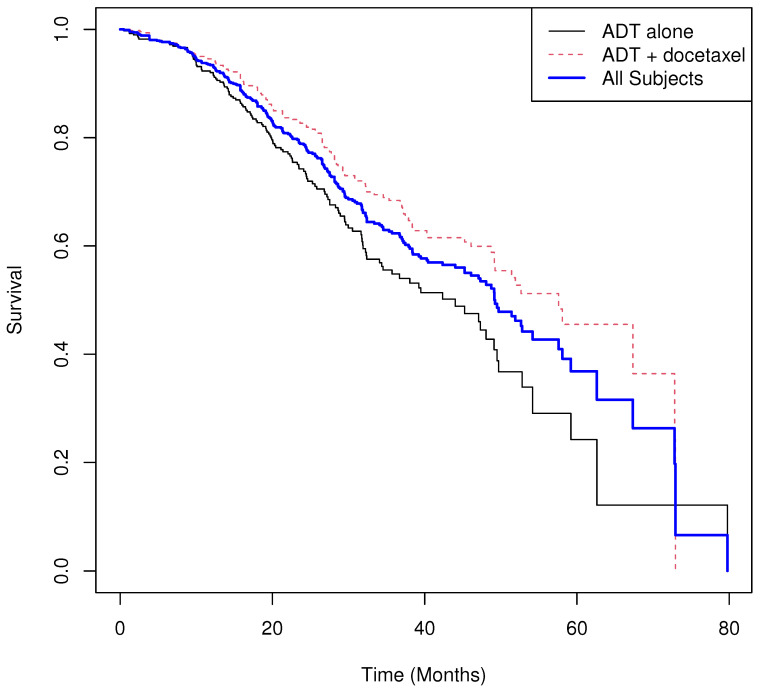
Real World Example 1: Estimated survival curves of ADT plus docetaxel versus those receiving ADT and combined survival curve estimate.

**Figure 6 cancers-17-03750-f006:**
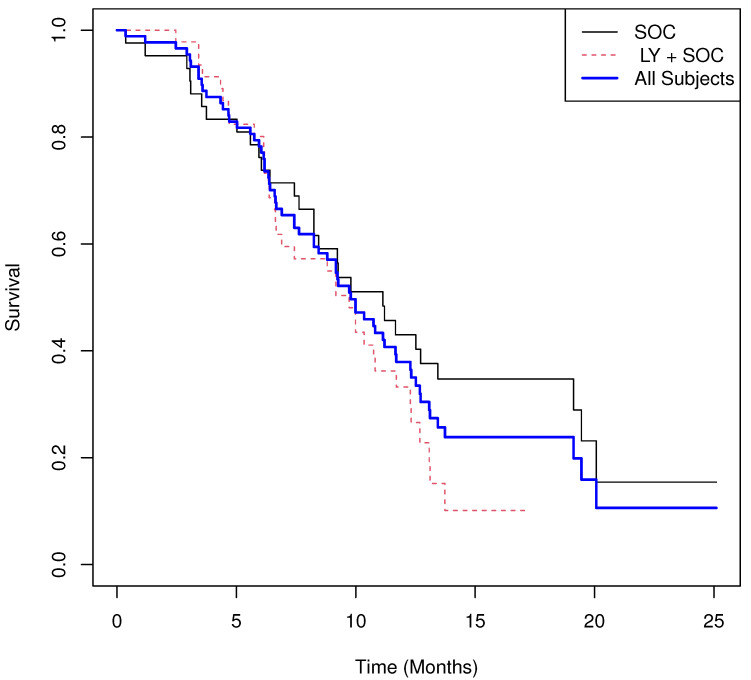
Real World Example 2: Estimated survival curves of LY + SOC versus those receiving SOC and combined survival curve estimate.

**Figure 7 cancers-17-03750-f007:**
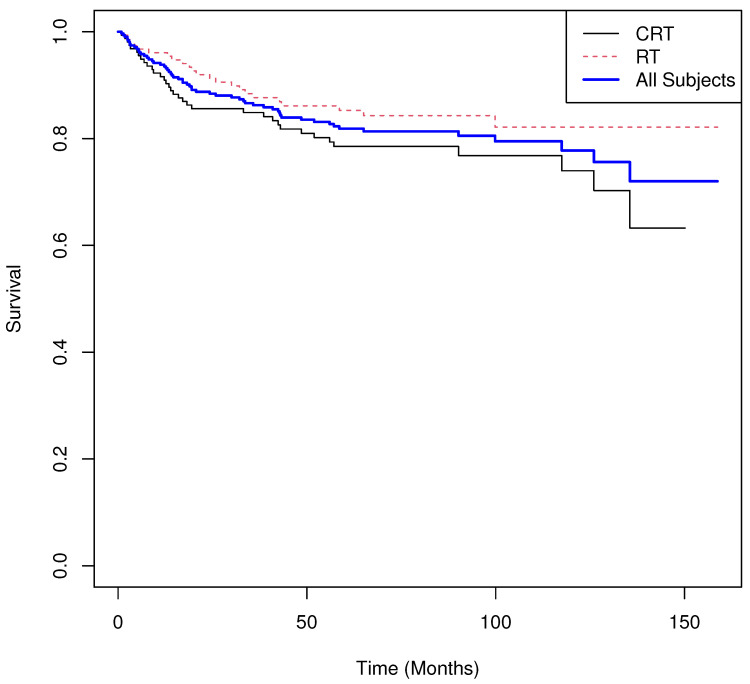
Real World Example 3: Estimated recurrence-free survival of CRT versus those receivin gRT and combined recurrence-free survival curve estimate.

**Table 1 cancers-17-03750-t001:** Simulation rejection proportions for permutation and logrank tests under varying $\left(\sigma_{0},\sigma_{c},\theta_{0}\right)$: Part I.

$\sigma_{0}$	$\sigma_{c}$	$\theta_{0}$	Permutation	Logrank
0.5	0.5	0.5	0.055	0.058
1.0	0.5	0.5	0.441	0.449
1.5	0.5	0.5	0.827	0.827
2.0	0.5	0.5	0.949	0.952
0.5	1.0	0.5	0.344	0.369
1.0	1.0	0.5	0.148	0.148
1.5	1.0	0.5	0.615	0.621
2.0	1.0	0.5	0.895	0.900
0.5	1.5	0.5	0.614	0.654
1.0	1.5	0.5	0.052	0.053
1.5	1.5	0.5	0.412	0.420
2.0	1.5	0.5	0.796	0.800
0.5	0.5	1.0	0.888	0.888
1.0	0.5	1.0	0.048	0.049
1.5	0.5	1.0	0.340	0.345
2.0	0.5	1.0	0.680	0.692
0.5	1.0	1.0	0.958	0.959
1.0	1.0	1.0	0.060	0.063
1.5	1.0	1.0	0.443	0.443
2.0	1.0	1.0	0.882	0.884
0.5	1.5	1.0	0.982	0.982
1.0	1.5	1.0	0.048	0.047
1.5	1.5	1.0	0.567	0.566
2.0	1.5	1.0	0.934	0.941

**Table 2 cancers-17-03750-t002:** Simulation rejection proportions for permutation and logrank tests under varying $\left(\sigma_{0},\sigma_{c},\theta_{0}\right)$: Part II.

$\sigma_{0}$	$\sigma_{c}$	$\theta_{0}$	Permutation	Logrank
0.5	0.5	1.5	0.996	0.996
1.0	0.5	1.5	0.215	0.221
1.5	0.5	1.5	0.100	0.102
2.0	0.5	1.5	0.352	0.355
0.5	1.0	1.5	0.999	0.999
1.0	1.0	1.5	0.111	0.109
1.5	1.0	1.5	0.278	0.277
2.0	1.0	1.5	0.749	0.749
0.5	1.5	1.5	0.997	0.997
1.0	1.5	1.5	0.054	0.057
1.5	1.5	1.5	0.472	0.484
2.0	1.5	1.5	0.931	0.936
0.5	0.5	2.0	1.000	1.000
1.0	0.5	2.0	0.526	0.526
1.5	0.5	2.0	0.042	0.047
2.0	0.5	2.0	0.209	0.215
0.5	1.0	2.0	1.000	1.000
1.0	1.0	2.0	0.237	0.241
1.5	1.0	2.0	0.176	0.181
2.0	1.0	2.0	0.658	0.665
0.5	1.5	2.0	1.000	1.000
1.0	1.5	2.0	0.108	0.110
1.5	1.5	2.0	0.401	0.412
2.0	1.5	2.0	0.887	0.893

## Data Availability

The data utilized to illustrate this work is publicly available.

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
