# Peer review of "An Alternative Treatment Effect Measure for Time-to-Event Oncology Randomized Trials"

_cancers, 2025, doi:10.3390/cancers17233750_

Round 1
Reviewer 1 Report
Comments and Suggestions for Authors
Thank you for the opportunity to review the manuscript titled “An Alternative Treatment Effect Measure for Time-to-Event Oncology Randomized Trials”.
This paper presents a new survival analysis measure, the univariate martingale residual (UMR), for comparing treatments in phase III oncology trials. The manuscript addresses important limitations of traditional methods, such as hazard ratios and log-rank tests, which rely on assumptions such as proportional hazards and large sample sizes that may not hold in complex study settings. Overall, the manuscript represents a timely and methodologically relevant contribution to survival analysis in oncology trials.
Comments:
1. The scenarios are described clearly, and the authors conclude with a reasonable summary of the main findings, particularly the establishment of the mean univariate martingale residual (UMR) as a robust and interpretable measure of treatment effect. However, several points could be addressed to strengthen this section:
- Clarification of practical implications: Although the methodological advantages of UMR over conventional hazard ratios are described, the conclusion could have been discussed more explicitly in terms of how this metric can be applied in practice by clinical researchers and trial participants.
- Acknowledgement of limitations: The conclusion emphasises the robustness of UMR, but does not address potential constraints such as data quality dependency, interpretation problems in small samples, or situations where UMR may not provide additional insight.
- Future perspectives: Including a brief look at future applications – such as validation in different clinical settings, comparison with alternative survival metrics, or integration into study design.
2. The list of references is generally up-to-date, with the majority of cited literature published between 2020 and 2025 (33 of 42 references), a smaller portion from 2015 to 2019 (only 2 refs), and a few older sources from before 2014 (7 refs). This suggests that the manuscript draws heavily on recent and relevant studies, which strengthens its timeliness and scientific relevance. However, authors may consider reviewing older references to ensure that they remain as relevant as possible and replacing them with more recent studies where possible.
Reviewer 2 Report
Comments and Suggestions for Authors
I really enjoyed reading this paper and thought the idea was clever, thoughtful, and well-presented overall.
Some questions and comments:
- Given the simulation performance demonstrating quite a bit of comparability with the log-rank test and the form of the statistic, I wondered if this statistic would be most powerful under proportional hazards like the log-rank statistic. Some work to consider under what settings this statistic is most powerful would be a very useful addition.
- Can the statistic be easily estimated using existing software?
- The scale of the statistic feels modest as presented here. 0.142 and 0.089 excess deaths (for example) feel modest for the significant p-values. Thoughts on scaling of the statistic by per 10 or per 100 by multiplying?
- The formulation of excess deaths compared to the combined curve estimates feels awkward. The combined curve doesn't really represent any particular patients in a randomized trial since patients are either in one or the other curve. Is there a variation on the statistic (say subtracted) that would give a comparison of the two arms against each other (excess deaths of A versus B instead of excess deaths versus the combo curve)?
Round 2
Reviewer 2 Report
Comments and Suggestions for Authors
If the proposed statistic is most powerful under PH, part of the motivation for the statistic seems to be lost. In both the abstract and background it is noted that log-rank tests can perform more poorly under non-PH settings. With the limited presentation here, it looks like the proposed method and log-rank test could have very similar properties. In which case, the maybe real benefit is in an effect size that can be estimated in the setting of 0 events in one arm or very, very small numbers of events? Other than that limited setting, it is not clear when the proposed statistic would yield a meaningful difference from Cox regression (since Cox-regression and log-rank tests have similar performance in most settings).
Without additional data presented, I think the motivation that this addresses non-PH settings should be toned down.
Author Response
Critique:
If the proposed statistic is most powerful under PH, part of the motivation for the statistic seems to be lost. In both the abstract and background it is noted that log-rank tests can perform more poorly under non-PH settings. With the limited presentation here, it looks like the proposed method and log-rank test could have very similar properties. In which case, the maybe real benefit is in an effect size that can be estimated in the setting of 0 events in one arm or very, very small numbers of events? Other than that limited setting, it is not clear when the proposed statistic would yield a meaningful difference from Cox regression (since Cox-regression and log-rank tests have similar performance in most settings).
Response: We agree with your assessment regarding power. However, the UMR based test is exact and does not rely on asymptotic assumptions and provides unbiased estimates of treatment effects regardless of proportional hazards assumptions being met. We have rewritten the abstract and conclusion to focus more these points.